# Impact of Mechanical Ventilation and Renal Replacement Therapy on Clinical Outcomes among Critically Ill COVID-19 Patients

**DOI:** 10.3390/reports6030031

**Published:** 2023-07-07

**Authors:** Anees A. Sindi

**Affiliations:** Department of Anesthesia and Critical Care, King Abdulaziz University Hospital, Faculty of Medicine, King Abdulaziz University, Jeddah 21589, Saudi Arabia; asindi2@kau.edu.sa

**Keywords:** COVID-19, chronic kidney disease, continuous renal replacement therapy (CRRT)

## Abstract

Background: Poor clinical outcomes in patients with severe COVID-19 occur due to many factors that require elucidation. The objective of this study was to describe the outcomes of critically ill patients with COVID-19 and identify the risk factors for mortality. Materials and Methods: The study was a single-centered cross-sectional, observational study involving COVID-19 patients admitted to the intensive care units (ICUs) of a tertiary care hospital in the Middle East and North Africa (MENA) region. The patients were admitted between 1 March and 31 December 2020. Logistic regression analysis was used to identify risk factors for mortality. Results: Of 107 patients admitted, 98 (91.6%) were ≥40 years old and 84 (78.5%) were males. The mean oxygen saturation at admission was 79.6 ± 12.6%, the duration of ICU stay was 13.0 ± 11.6 days, and 65 (60.7%) of the patients received mechanical ventilation. Major comorbidities included hypertension (57%), diabetes (56%), and chronic kidney disease (CKD) (15.5%). The overall mortality rate was 51.4%; this was higher in patients who received mechanical ventilation (60 vs. 38.1%; *p* = 0.03) and those with co-morbid hypertension (60.7 vs. 39.1%; *p* = 0.03). Risk factors for mortality were: need for mechanical ventilation agent of record adjusted Odds ratio (aOR) 4.4 (1.6–12.6), co-morbid hypertension aOR 5.8 (1.6–21.1), having CKD aOR 5.4 (1.2–25.6) and receiving renal replacement therapy aOR 4.3 (1.4–13.0). Conclusions: The use of mechanical ventilation or renal replacement therapy among critically ill COVID-19 patients could potentially predict worse outcomes. Patients with existing hypertension or CKD may carry a higher mortality risk.

## 1. Introduction

The Coronavirus disease of 2019 (COVID-19), caused by the novel coronavirus—severe acute respiratory syndrome coronavirus 2 (SARS-CoV-2)—became a pandemic with rapidly fluctuating epicenters, from China to the United States and from the United Kingdom to India [1]. The broad family of viruses known as coronaviruses (CoV) can cause illnesses ranging from the common cold to more serious conditions like Middle East Respiratory Syndrome (MERS and SARS). A new coronavirus strain that has not been discovered in people before is known as a novel coronavirus (nCoV). Since coronaviruses are zoonotic, they can spread from people to animals. In Wuhan, Hubei, China, a number of pneumonia cases with unknown causes surfaced in December 2019. The clinical presentations were strikingly similar to viral pneumonia. Lower respiratory tract sample deep sequencing study revealed a novel coronavirus, designated 2019 novel coronavirus (2019-nCoV). The World Health Organisation (WHO) designated COVID-19 to be a pandemic in February 2020, and by 19 December 2021, the WHO Situation Report had confirmed that COVID-19 had spread globally. By 28 October 2020, the ministry of health (MOH) had reported a total of 345,631 confirmed cases with a case fatality rate of 0.86%. The first confirmed case of COVID-19 was originally reported in Saudi Arabia on 2 March 2020 [2]. Although most patients affected with SARS-CoV-2 are asymptomatic or developed mild to moderate symptoms such as respiratory problems, fall in oxygen saturation in room air, ageusia, sore throat, and fever [3,4], approximately 5% of the patients go on to develop critical COVID-19 characterized by acute respiratory distress syndrome (ARDS) and/or sepsis, requiring Intensive Care Unit (ICU) admission and mechanical ventilation [5]. According to the Chinese Centre for Illness Control and Prevention’s summary report, cancer (5.6%), diabetes (7.3%), chronic respiratory illness (6.3%), hypertension (6.0%), and cardiovascular disease (10.5% of all fatalities) were the main causes of mortality. The senior population aged 70–79 years was among the highest risk groups in this analysis, with a documented case fatality rate of 8%. Additionally, a greater death risk of 49% was noted for the seriously sick subgroup population in the aforementioned analysis [6]. As the pathophysiology of the disease started to unfold, it became evident that the lungs are not the only end-organs that are affected by the disease but other end-organs, such as the kidneys and heart, are also affected [6]. Therefore, multiple organ failure is common in patients presenting with critical COVID-19 [3]. Additionally, the illness is linked to high fatality rates among seriously ill people globally. Acute kidney injury (AKI), multiorgan failure, and acute respiratory distress syndrome (ARDS) are just a few of the problems that have been seen in critically ill COVID-19 patients. Many of these patients have also needed mechanical ventilation [7]. It was found that critically ill COVID-19 patients encounter various kidney-related complications such as electrolyte derangements, acute kidney injury (AKI), and glomerular disease [7]. Although some studies claimed that SARS-CoV-2 infection does not cause AKI and that AKI is uncommon in COVID-19, others claimed that renal replacement therapy (RRT) was necessary in 16–23% of intensive care unit (ICU) or mechanically ventilated patients and that the incidence of AKI among COVID-19 infected patients ranges from 0.5% to 50% among non-survivors. This controversy may be related to the variety of symptoms seen in the early stages of the COVID-19 epidemic and the absence of conclusive diagnostic resources [8]. In Wuhan City, China, it was described the incidence of AKI attributable to COVID-19 and accompanying mortality in 701 hospitalised patients from a single medical centre. Although AKI was 5.1% in frequency, there was a dose-dependent correlation between the severity of AKI and mortality. On admission, there was a 44% incidence of proteinuria and a 27% incidence of haematuria. It is volatile whether the AKI or underlying chronic kidney disease (CKD) caused the proteinuria and haematuria [3]. Patients who require dialysis and kidney transplant recipients have previously been recognised as a subgroup at higher risk for poor outcomes. These patients frequently exhibit unique clinical traits that provide an extra burden. Chronic kidney disease (CKD), with a clear correlation between the degree of dysfunction and death rate, has only lately been shown to be a major risk factor for COVID-19 mortality. A significant prevalence of renal problems, such as severe proteinuria, haematuria, and raised serum creatinine and blood urea nitrogen levels, could result from COVID-19 infection [9].

The death rate from pneumonia appears to be 14–16 times greater in CKD patients than in the general population, and CKD is linked to an increased risk of pneumonia. The two main predictors of in-hospital death were advanced age and chronic renal disease. Acute renal injury, ARDS, the need for mechanical ventilation, and ICU admission were all more common in non-survivors than in survivors. Due to a shortage of personal protective equipment and restricted transportation during the lockdown, COVID-19 appears to be a concern to the health of CKD patients in terms of follow-up and medication acquisition [10].

In fact, the most common risk factor responsible for the mortality of such patients is chronic kidney disease (CKD) and the risk of death increases with a higher stage of CKD or patients with kidney failure receiving dialysis [11,12]. A study reported 40% mortality among ICU-admitted COVID-19 patients receiving dialysis (*n* = 197). However, there was no significant difference in mortality rates at three months post-discharge in such patients [13]. Two explanations have been proposed to explain the onset of AKI in SARS-CoV-2-infected patients: a direct cytotoxic effect on kidney tissues and tissue hypoxia in the presence of large cytokine production as well as the systemic effects of the virus [14].

An updated systematic review conducted by Dorjee et al. showed an increased risk of severe COVID–19 infection among patients with CKD with a relative risk of 1.6 (95% CI 1.3–1.9) [15]. Another systematic review concluded that patients with CKD were more likely to have worse outcomes from COVID–19 compared to patients without CKD [16]. Several other previous studies evaluated the effect of CKD on the severity of COVID–19 infection, showing a higher odds ratio for severe COVID–19 infection ranging from 2.1 (95% CI 1.2–3.8) to 3.6 (95% CI 2.2–5.8) (12–14). On the contrary, Forest et al. reported that concomitant renal failure in critical COVID-2019 was not a significant predictor of death (odds ratio 2.3; *p* = 0.057) [17].

Renal replacement therapy (RRT), either continuous or intermittent, is indicated for critically ill COVID-19 patients presenting with AKI. RRT improves hemodynamic attributes and reduces the requirement for mechanical ventilation in such patients. Previous studies have indicated that a substantial percentage of COVID-19 patients exhibit AKI and require RRT or conventional haemodialysis [18]. One study showed that 36.6% of critically ill COVID-19 patients developed AKI [19], while another study reported the percentage to range between 24% and 35% for which RRT is required in 15% to 35% of patients presenting with AKI [11]. These findings suggest RRT plays a role in reducing the risk of CKD in AKI patients. From this perspective, it could be assumed that renal function in critically ill COVID-19 patients with CKD would be compromised too, requiring RRT. The indications for RRT in COVID-19 patients include cytokine storm, uremic pericarditis, hypervolemia, and mechanical ventilation (MV). This is because volume overload often reduces the efficacy of ventilator support [20]. Based on these findings, the National Institutes of Health (NIH) COVID-19 Treatment panel recommends the use of RRT in COVID-19 patients with AKI while prolonged intermittent RRT (IRRT) or intermittent dialysis is recommended in the absence of AKI [21]. Continuous RRT (CRRT) is solicited in preference to IRRT because they do not require nursing intervention reducing the risk of cross-infection. The dosing in CRRT is more easily optimized than in IRRT. Patients with stage 3 AKI (defined as either an increase in creatinine levels of a 3-fold from baseline or creatinine 4.0 mg/dL or a decrease in the amount of urine output:0.3 mL/kg/h for 24 h or anuria for 12 h) who are hospitalized in the ICU should receive CRRT, according to COVID-19 patients who have AKI. The World Health Organisation (WHO) advice issued a statement in which it recommended referring patients with refractory hypoxemia in cases of severe ARDS [22].

Despite lung-protective ventilation in environments where extracorporeal membrane oxygenation (ECMO) is a specialty, CRRT techniques can be employed in conjunction with ECMO [23]. However, the evidence does not indicate that CRRT is superior to intermittent dialysis in COVID-19 patients presenting with AKI in terms of renal recovery or mortality reduction, but it certainly ensures stable hemodynamics that could prevent cardio-respiratory failure. Respiratory failure and the requirement for mechanical ventilation have been distinguishing symptoms of the SARS-CoV2 epidemic, stretching the capacity of the healthcare system’s staff and equipment. According to early estimates, the mortality rate for people who are mechanically ventilated might be unusually high, reaching 65–81% and more recent figures indicate that mortality is nearing “traditional” ARDS of any origin. Patients with COVID-19 respiratory failure are more likely to develop AKI after starting MV. These individuals had a significant death rate and a high need for dialysis while on MV. All of these patients had noticeably increased CRP and d-dimer levels. Possible causes of hypotension, sepsis, and eventual AKI include oxygen toxicity, capillary endothelial injury from MV, and capillary endothelial damage from oxygen exposure. Iatrogenic (drug-related) causes and direct viral cytotoxicity are two other potential pathways causing AKI [24].

We hypothesized that CKD patients affected with COVID-19 would be prone to deterioration in renal function which might further predispose the risk of respiratory failure, while CRRT might reduce such risks. Hence, with this intent, a retrospective analysis was conducted to evaluate the role of CRRT on clinical outcomes among critically ill COVID-19 patients with CKD.

## 2. Materials and Methods

This study was conducted as a single-centred cross-sectional, retrospective, and observational study involving a tertiary care centre in the MENA region. All patients (*n* = 107) were admitted to the ICU between 1 March and 31 December 2020. The patients who did not require ICU admission and those with incomplete health records were excluded from the study. The major inclusion criterion for the study was “ICU admission for COVID-19” and aged 18 years or older.

The demographic profile that was obtained included age, gender, comorbidities, body mass index (BMI), mean oxygen saturation, length of ICU stay, number of patients on mechanical ventilation, patients with CKD, patients requiring CRRT, and mortality at discharge. Confirmatory diagnosis of CKD was based on electronic health records and abnormal serum creatinine levels at baseline and on arrival at the hospital. As per the treatment protocol, CKD patients who required RRT were initiated on CRRT. The demographic and clinical parameters were compared between CKD and non-CKD patients.

The results were reported as descriptive statistics and inferential statistics. The descriptive statistics included frequency (proportions) as well as means ± standard deviation (SD). The inferential statistics include Chi-square test for comparison of categorical variables and Students t-test for comparing quantitative variables. Students t-tests were used for comparing quantitative variables because they conformed to a normal distribution as depicted by Shapiro–Wilk tests. All statistical tests of inference were evaluated at the 0.05 level of significance (two-sided test).

The study was conducted after obtaining appropriate permission and approval from the Institutional Review Board (IRB) of the hospital. The identity of each patient was kept confidential while reporting the results of this study.

## 3. Results

A total of 107 patients were admitted during the period; 98 (91.6%) of them were 40 years old or over with a mean age of 57.6 ± 12.8 years. The majority of the patients were males, 84 (78.5%), and non-Saudi nationals, 96 (89.7%). The demographic and clinical characteristics of the patients are shown in Table 1. Some of the patients had co-morbidities like diabetes mellitus, 60 (56.1%), hypertension, 61 (57.0%), heart failure, 13 (12.1%), or CKD, 17 (15.9). A substantial portion of the patients were overweight and obese 71 (66.3%). The mean oxygen saturation at admission was 79.6 ± 12.6%. The mean duration of ICU stay was 13.0 ± 11.6 days, and 65 (60.7%) of the patients required and received mechanical ventilation. However, there was no significant difference in the number of patients receiving mechanical ventilation among CKD (15.4% vs. 16.7%), and non-CKD patients (84.6% vs. 83.3%; *p* = 0.86).

Overall, 55 (51.4%) of the patients died. Table 2 shows the distribution of mortality and survival of the patients according to the sociodemographic and clinical characteristics of the subjects. Overall, mortality rate among the patients did not significantly differ by age, gender, or nationality (Table 2). Mortality rate among patients who received mechanical ventilation was 60% compared to 38.1% among those who did not receive it (*p* = 0.03). Also, patients with co-morbid hypertension were more likely to die than those without hypertension (60.7% vs. 39.1%; *p* = 0.03). A higher proportion of patients who received renal replacement therapy were more likely to die compared to those who did not receive it (55.6% vs. 12.5%), but the difference was not statistically significant (*p* = 0.06).

Multivariable logistic regression analysis of risk factors of mortality among critically ill COVID-19 patients are as shown in Table 3. Overall, receiving mechanical ventilation aOR 4.4 (95% C.I. 1.6–12.6), having co-morbid hypertension aOR 5.8 (95% C.I. 1.6–21.1), having CKD aOR 5.4 (95% C.I. 1.2–25.6), and receiving CRRT aOR 4.3 (95% C.I. 1.4–13.0) were independent predictors of mortality among critically ill COVID-19 patients. Having other co-morbid diseases like diabetes mellitus, heart failure, or level of oxygen saturation on admission and duration of ICU stay were not predictors of mortality among the patients.

## 4. Discussion

The prognosis of critically ill COVID-19 patients is still a subject of debate and dilemma. The mortality and morbidity associated with COVID-19 infection are primarily attributed to ARDS and resultant sepsis [6,21,22,23,24,25]. The risk of end-organ failure is high in critically ill COVID-19 patients and is primarily attributed to multi-organ inflammation [25,26]. In this regard, Morrow et al. elucidated the importance of cardio-renal homeostasis in ensuring hemodynamic stability and treatment outcomes in COVID-19 patients [27]. Health professionals are seeking to understand the odds of mortality as a function of end-organ failure in patients affected with severe or critical COVID-19 [25]. Therefore, it is necessary to preserve or improve renal function in such patients [26,27].

The American Society of Nephrology (ASN) specialists have endorsed CRRT as one of the recommended RRT in COVID-19 patients with severe AKI [28].

Different studies [6,21,25,27] have reported the impact of acute renal injury (ARI) and CRRT on mortality outcomes in COVID-19 patients but the evidence on CKD on mortality outcomes in COVID-19 remains inconclusive; particularly, from the perspective of CRRT use. Some of the CKD patients suffering from critically ill COVID-19 went on to require CRRT, unlike their non-CKD counterparts, because the former have baseline poor renal function compared to the latter. In this study, we found that over half of critically ill COVID-19 patients admitted in the ICU in our setting died. Also, we report that the predictors for mortality in this population includes, receiving mechanical ventilation, having co-morbid hypertension, having CKD, and receiving CRRT. When it comes to a staffing and equipment shortfall, CRRTs present special difficulties in healthcare facilities. While providing CRRT, there is a substantial danger of exposure. Due to clustering, the risk of infection is also considered in the outpatient dialysis scenario. When dealing with COVID-19 patients, outpatient dialysis is fraught with staffing, transportation, and exposure risks. Our findings suggest that the mortality rate among critically ill COVID-19 patients with CKD was higher than non-CKD patients. It is worth noting that most of the CKD patients received CRRT, and receiving this remained an independent predictor for mortality.

Our findings are consistent with those of various studies [29] which report that CKD increases the severity of COVID-19 infection. Hittesdorf et al. reported a mortality rate of 40% among patients who were admitted with COVID-19 requiring CRRT, mechanical ventilation, and ICU care as compared to the mortality rate of 23.9% for non-CRRT patients during hospitalization and after 90 days [30]. The finding raises the dilemma of whether CRRT increased the risk of mortality in COVID-19 patients, or if the patients in the CRRT group were simply more critical, as they required mechanical ventilation. The second assumption is supported by the evidence that CRRT provides slower solute and volume clearance per unit time that prevents the risk of volume overload, hypovolemia, and resultant myocardial infarction [31]. CRRT is indicated for managing volume overload, acid-base abnormalities, azotaemia, uraemia, and electrolyte imbalance. Higher mortality rates could be attributed to the age of the patient, comorbidities, and severity of the disease and deterioration in renal function. Thus, the prognosis of critically ill COVID-19 patients might also depend on other factors, aside from renal function [12,25,26,27,28,29,30,31,32]. This study also showed that receiving renal replacement therapy was associated with higher odds of mortality at hospital discharge, which supports the findings of Forest et al. [33]. Similarly, Burke et al., also found that need for continuous veno-venous hemofiltration was an independent risk factor for mortality among critically ill COVID-19 patients [34].

Although more than half of the patients required and received mechanical ventilation, we found no association between CKD and the use of mechanical ventilation; we that receiving mechanical ventilation during the admission was an independent risk factor for mortality. Our findings are consistent with those of other studies. Burke et al. reported that the overall hospital mortality rate of critically ill patients with COVID-19 was 36%, versus 56% for those requiring renal replacement therapy, and 68% for those with both mechanical ventilation and renal replacement therapy [35]. Nandy et al. found that the odds ratio for adverse events was higher in CKD patients compared to non-CKD patients (OR 5.32) [36]. Similarly, the rates of serious adverse events were higher in patients with co-morbid hypertension, diabetes mellitus, and chronic obstructive pulmonary disease [37]. Thus, it could be contended that the presence of co-morbidities and receiving interventions like haemodialysis and mechanical ventilation increases the risk of mortality in critically ill COVID-19 patients; identifying and addressing these risks may improve survival.

COVID-19 patients are at risk of poor renal function because the release of pro-inflammatory mediators could lead to acute tubular necrosis, especially in those with sepsis and shock [21]. The evidence suggests that CRRT controls the potential damage caused by cytokine storms, capillary leaks, and dysregulated immune responses in sepsis patients. The capillary leaks lead to organ dysfunction due to volume overload. Such attributes increase the need for ventilator support [38]. Thus, CRRT, by addressing volume overload and sepsis, increases the efficacy of ventilatory support [39]. The importance of CRRT is further elucidated by Paramitha et al. who highlighted how drug removal in COVID-19 patients with CKD depends on protein binding [40]. Drugs that are less protein bound are more easily cleared compared to those which remain protein bound. Therefore, CRRT could reduce the risk of secondary infection in COVID-19 patients.

Although this study elucidated the predictors for mortality in critically ill COVID-19 patients requiring ICU admissions, it had various limitations. First, this was a retrospective observational single-centred study. Therefore, we are unable to make any causal deductions and our finding may not be generalizable to the whole country or region. Second, we did not assess the pro-inflammatory cytokine levels in order to evaluate the efficacy of CRRT in addressing this clinical challenge in COVID-19 patients. Third, all measurements in this study were performed during the routine care of the admitted patients rather than in accordance with a clinical trial protocol. The advantage therefore is that it reflected real-world scenarios of the management of these patients. However, during the period of care of these patients, there were some changes in the management protocol for severe COVID-19 such as the introduction of remdesivir and dexamethasone which were not given to all the patients in this cohort. The impact of these therapies on renal function and co-morbid conditions associated with severe COVID-19 is not clear. Finally, the study was biased toward gender because there were more male patients than female. Despite these limitations, our paper adds to the extant literature on the predictors of mortality in patients with severe COVID-19.

## 5. Conclusions

Our study showed that there was a high mortality rate among critically ill COVID-19 patients requiring ICU admissions. Also, this is driven by co-morbid hypertension and CKD. Also, a high mortality rate was observed among patients having the need for mechanical ventilation and renal replacement therapy. Our study contributes to the body of knowledge on the factors that influence mortality in individuals with severe COVID-19. Strategies to address these identified risk factors in the study population is recommended.

## Figures and Tables

**Table 1 reports-06-00031-t001:** Socio-demographic and clinical characteristics of the study participants.

Variable	*n* (%)
**Age (years)**	
<40	9 (8.4)
≥40	98 (91.6)
Mean ± SD	57.6 ± 12.8
**Sex**	
Female	23 (21.5)
Male	84 (78.5)
**Nationality**	
Saudi	11 (10.3)
Non-Saudi	96 (89.7)
**Co-morbidities**	
Diabetes mellitus	
No	47 (43.9)
Yes	60 (56.1)
**Hypertension**	
No	46 (43.0)
Yes	61 (57.0)
**Heart failure**	
No	94 (87.9)
Yes	13 (12.1)
**Chronic kidney disease**	
No	90 (84.1)
Yes	17 (15.9)
**Body mass index (kg/m^2^)**	
Normal	36 (33.7)
Overweight	44 (41.1)
Obese	27 (25.2)
**Mechanical ventilation**	
No	42 (39.3)
Yes	65 (60.7)
**Mortality**	
No	52 (48.6)
Yes	55 (51.4)
**Oxygen saturation (%)**	
Mean ± SD	79.6 ± 12.6
**Days of ICU admission (days)**	
Mean ± SD	13.0 ± 11.6

SD = Standard deviation; ICU = Intensive Care Unit.

**Table 2 reports-06-00031-t002:** Mortality among of Critically ill COVID-19 patients (*n* = 107).

Variable	Alive *n* (%)	Death *n* (%)	*p*-Value
Total	52	55	
**Age** (**years**)			0.66
<40	5 (55.6)	4 (44.4)	
≥40	47 (48.0)	51 (52.0)	
**Sex**			0.31
Female	9 (39.1)	14 (60.9)	
Male	43 (51.2)	41 (48.5)	
**Nationality**			0.29
Saudi	45 (46.9)	51 (53.1)	
Non-Saudi	7 (63.6)	4 (36.4)	
**Body mass index** (**kg/m^2^**)			0.45
Normal	19 (52.8)	17 (47.2)	
Overweight	21 (47.7)	23 (52.3)	
Obese	12 (44.4)	15 (55.6)	
**Mechanical ventilation**			0.03
No	26 (61.9)	16 (38.1)	
Yes	26 (40.0)	39 (60.0)	
**Diabetes mellitus**			0.74
No	22 (46.8)	25 (53.2)	
Yes	30 (50.0)	30 (50.0)	
**Hypertension**			0.03
No	28 (60.9)	18 (39.1)	
Yes	24 (39.3)	37 (60.7)	
**Heart failure**			0.85
No	46 (48.9)	48 (51.1)	
Yes	6 (46.2)	7 (53.8)	
**Chronic kidney disease**			0.15
No	41 (45.6)	49 (54.4)	
Yes	11 (64.7)	6 (35.3)	
**CRRT** (***n* = 17**)			0.06
No	7 (87.5)	1 (12.5)	
Yes	4 (44.4)	5 (55.6)	
**Oxygen saturation** (**%**)			
Mean ± SD	79.10 ± 10.1	76.1 ± 14.5	0.06 **
**Days of ICU admission**			
Mean ± SD	10.4 ± 13.4	15.4 ± 13.6	0.23 **

CRRT = continuous renal replacement therapy; ICU = Intensive Care Unit; ** *p*-value based on ANOVA; all other *p*-values. Based on Chi-square test.

**Table 3 reports-06-00031-t003:** Risk factors of mortality among of critically ill COVID-19 patients.

Variable	Mortality *n* (%)	Crude OR (95% C.I.)	Adjusted OR (95% C.I.)	Adjusted *p*-Value
**Age** (**years**)				
<40	4 (44.4)	1	1	0.73
≥40	51 (52.0)	1.4 (0.3–5.4)	0.7 (0.1–4.5)	
**Sex**				
Female	14 (60.9)	1.6 (0.6–4.2)	1.2 (0.3–54.4)	0.78
Male	41 (48.5)	1	1	
**Nationality**				
Saudi	51 (53.1)	1.9 (0.5–7.2)	1.8 (0.4–9.0)	0.45
Non-Saudi	4 (36.4)	1	1	
**Body mass index** (**kg/m^2^**)				
Normal	17 (47.2)	1	1	0.26
Overweight	23 (52.3)	1.2 (0.5–3.0)	1.9 (0.6–5.8)	
Obese	15 (55.6)	1.4 (0.5–3.8)	1.1 (0.3–4.2)	
**Mechanical ventilation**				
No	16 (38.1)	1	1	0.90
Yes	39 (60.0)	2.4 (1.1–5.4)	4.4 (1.6–12.6)	
**Diabetes mellitus**				
No	25 (53.2)	1.1 (0.5–2.4)	0.5 (0.1–1.6)	0.005
Yes	30 (50.0)	1	1	
**Hypertension**				
No	18 (39.1)	1	1	0.21
Yes	37 (60.7)	2.4 (1.1–5.3)	5.8 (1.6–21.1)	
**Heart failure**				
No	48 (51.1)	1	1	0.007
Yes	7 (53.8)	1.1 (0.3–3.6)	1.5 (0.3–7.5)	
**Chronic kidney disease**				
No	49 (54.4)	2.2 (0.7–6.4)	1	0.59
Yes	6 (35.3)	1	5.4 (1.2–25.6)	0.03
**CRRT**				
No	1 (12.5)	1	1	
Yes	5 (55.6)	8.7 (0.7–103.8)	4.3 (1.4–13.0)	0.009
**Days of ICU admission**		1.0 (0.9–1.1)	1.0 (0.9–1.1)	0.10
**Oxygen saturation**		1.0 (0.9–1.1)	1.0 (0.9–1.1)	0.38

CRRT = continuous renal replacement therapy; ICU = Intensive Care Unit; OR = Odds ratio; 95% C.I. = 95% Confidence Interval.

## Data Availability

The data presented in this study are available on request from the corresponding author.

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
