# Peer review of "Impact of Mechanical Ventilation and Renal Replacement Therapy on Clinical Outcomes among Critically Ill COVID-19 Patients"

_reports, 2023, doi:10.3390/reports6030031_

Round 1

Reviewer 1 Report

Dear Authors,

The introduction is too short. 3 years after the pandemic, accumulated data allow a broad approach to the problem presented. The retrospective study carried out over a period of 9 months is brief. The analyzed parameters are simplistic and focused more on demographic data and comorbidities. Please complete with the criteria that were the basis of the stated notions - heart failure, chronic kidney disease... etc.

Author Response

Thank you for your email requesting for revision of the above-mentioned manuscript. Please find the attached response to your comments 

Reviewer 2 Report

The paper is well done and well presented however little sounds really innovative, and that could be a motive for rejection...

A suggestion could be to highlight and give more details about the collaterals due to MV...

Author Response

Thank you for requesting revisions for the manuscript. Attached herewith is a detailed response addressing each of your comments.

Round 2

Reviewer 1 Report

Dear authors,

The additions made correspond to the requests.

Congratulations!

Author Response

Thank you for your valuable work on our manuscript, which helped us to improve the manuscript.

Reviewer 2 Report

Though there is no novelty this could be accepted 

Though there is no novelty this could be accepted

Author Response

(The authors gave the same response as above.)
